# Comparative analysis of surgical outcomes: Video-assisted thoracoscopic surgery versus open thoracotomy in organizing thoracic empyema management

Thanat Suwanarawat[1][☉], Rachata Deesang[1][☉], Sasikarn Tawanseree[1][☉],
Sikkawat Watcharasoonthonkit[1][☉], Somcharoen Saeteng[2]*, Sophon Siwachat[2],
Apichat Tantraworasin [iD][2,3]*

1 Faculty of Medicine, Chiang Mai University, Chiang Mai, Thailand, 2 Department of Surgery, Faculty of Medicine and Clinical Surgical Research Center, Chiang Mai University, Chiang Mai, Thailand, 3 Clinical Surgical Research Center, Chiang Mai University, Chiang Mai, Thailand

☉ These authors contributed equally to this work.
* tengearneae@gmail.com (SS); Apichat.t@cmu.ac.th (AT)

## Abstract

### Background

Surgical intervention is essential for managing organizing thoracic empyema, but the efficacy of Video-Assisted Thoracic Surgery (VATS) in this disease stage remains debated. This study aims to compare the surgical outcomes of VATS versus open thoracotomy (OT) in the management of organizing thoracic empyema.

### Materials and methods

This retrospective cohort study included 393 patients who underwent surgery for organizing thoracic empyema at Maharaj Nakorn Chiang Mai Hospital between January 1, 2012, and December 31, 2022, and were divided into VATS and OT groups. The primary outcomes were lung full expansion before discharge and at the 2-week follow-up. Secondary outcomes included intraoperative blood loss, duration of intensive care unit (ICU) stay, and postoperative pain scores. Data analysis was performed using multivariable regression analysis and propensity score matching.

### Results

In the propensity-matched cohort (212 patients), patient characteristics were balanced between the two groups. VATS was associated with a higher likelihood of full lung expansion at discharge (risk ratio: 1.21; 95% CI: 1.01 to 1.45compared to OT. There were no significant differences in postoperative pain scores, or full lung expansion at 2 weeks after discharge between the two groups. However, the VATS

**Funding:** The study was partially supported by a grant from the Chiang Mai University, Chiang Mai, Thailand (Grant number R68IN00016). The funders had no role in study design, data collection and analysis, decision to publish, or preparation of the manuscript. There was no additional external funding received for this study.

**Competing interests:** The authors have declared that no competing interests exist.

group showed a trend toward less intraoperative blood loss (mean difference: −34.20; 95%CI: −162.89 to 94.49).

## Conclusions

VATS offers advantages over OT in organizing thoracic empyema surgery, particularly in terms of improved lung expansion at discharge. Further studies with larger sample sizes are warranted to support these findings.

## Introduction

Thoracic empyema, a collection of pus in the pleural space, is caused by an inflammatory process resulting from infection. Its incidence is gradually increasing each year and is associated with significant morbidity and mortality [1]. Thoracic empyema is relatively common complication of pneumonia, and other causes may include lung abscesses, chest trauma, or surgical complications. According to the American Thoracic Society, thoracic empyema progresses through three stages: 1) Exudative stage, this early phase occurs within the first few days of infection, characterized by thin, non-loculated exudative fluid accumulation, 2) Fibrinopurulent stage, in this intermediate phase, pleural fluid becomes turbid, with the presence of fibrin deposits, and 3) Organizing stage, this chronic phase involves the organization of fibrinopurulent material into thicker pus, leading to the formation of a fibrin peel within the pleural cavity. Surgical treatment is crucial in managing organizing thoracic empyema due to the thick fibrin peel deposits on the visceral pleura, which can cause loculated pleural effusions and limit lung expansion. Currently, there are two main surgical approaches: Video-Assisted Thoracic Surgery (VATS) and open thoracotomy (OT). The choice between VATS and OT for the treatment of thoracic empyema is a topic of ongoing discussion in the medical field.

Over the past 5–10 years, VATS has gained popularity and emerged as a safe procedure for thoracic surgery. Several studies have reported the efficacy of VATS in treating multiloculated and chronic empyema, demonstrating benefits such as reduced duration of surgery, decreased pain, fewer complications, and shorter hospital stays compared to OT [1–3]. However, a high incidence of conversion from VATS to OT due to unsuccessful VATS has been reported, particularly in later stages of the disease. Despite its effectiveness in the early stages, limitations persist for VATS in the later stages of thoracic empyema [1,4]. Given the ongoing controversy and questions surrounding the outcomes of both surgical approaches, more comparative data is needed to provide reliable information for treatment considerations. This study aims to compare the surgical outcomes of VATS and OT in the management of organizing thoracic empyema.

## Materials and methods

This study was reviewed and approved by the Research Ethics Committee, Faculty of Medicine, Chiang Mai University, Chiang Mai, Thailand (No: 088/2566), with

a waiver for written informed consent due to its retrospective nature. This retrospective cohort study was conducted at Maharaj Nakorn Chiang Mai Hospital. Data were collected retrospectively from the medical records of all patients diagnosed with organizing phase thoracic empyema (stage III) who underwent pleural decortication via either VATS or OT between January 1, 2012, and December 31, 2022. The data was collected between April 1, 2023, and December 31, 2023. At our institution, VATS has been used for thoracic empyema since 2015. Prior to this period, OT was performed for all decortication cases. After 2015, VATS became the initial procedure for all thoracic empyema surgeries, including those in organizing phase. Both VATS and OT procedures were performed by a consistent team of thoracic surgeons, all of whom had experience with both techniques.

Inclusion criteria encompassed all adult patients over the age of 20 diagnosed with organizing phase thoracic empyema. Patients included in this study were those with stage III thoracic empyema who underwent surgical intervention. Due to the challenge of accurately determining empyema stage preoperatively, the decision for surgery was based on persistent sepsis or respiratory compromise despite appropriate medical therapy, in conjunction with radiological findings suggestive of organized empyema or pleural thickening. Definitive staging was established intraoperatively. Stage III empyema was defined by the presence of a thickened, fibrous pleural peel requiring decortication at surgery [5]. Patients who initially underwent VATS but were converted to OT due to a very thick organized peel covering the visceral pleura, resulting in incomplete decortication, or due to unexpected complications, were excluded from the primary analysis to minimize indication bias (per-protocol approach). However, to reflect real-world practice, a sensitivity analysis that included conversion cases was performed using both intention-to-treat and as-treated analytic approaches.

The collected data included patient characteristics (age, sex, underlying diseases, etc.), symptoms and their duration, previous treatments, stage of the disease, imaging findings, pathogens, surgical details, postoperative outcomes, and follow-up imaging.

## Surgical technique of video-assisted thoracoscopic surgery (VATS)

The VATS procedure was performed under general anesthesia with double-lumen endotracheal intubation. Patients were positioned in the contralateral decubitus position. At our hospital, a two-port VATS technique is typically employed. A 4 cm incision was made at the 5th intercostal space (ICS) along the anterior axillary line to create the utility port (Fig 1). The surgeon performed digital exploration by introducing the index finger into the thoracic cavity. A 30-degree rigid endoscope was used to examine the pleural cavity and to identify the site for the camera port, which was usually placed at the 7th

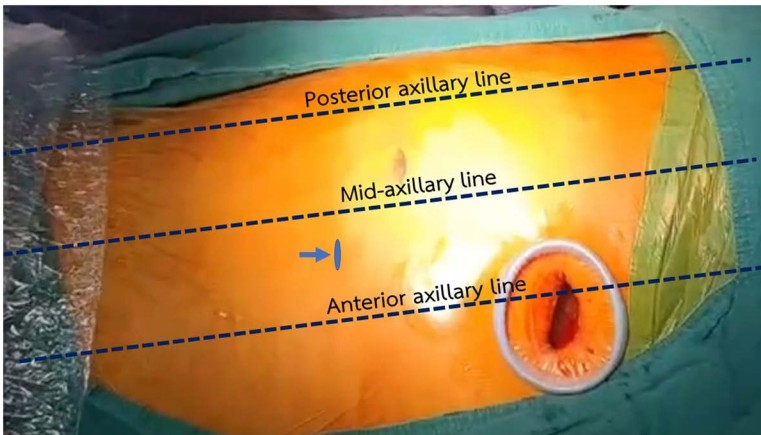

**Fig 1. Location of skin incisions for the utility port and camera port.**

or 8th ICS along the anterior to the mid-axillary line. Under endoscopic guidance, fibrin, loculated effusion, and septated effusion were removed via thoracoscopic suction devices and/or forceps sponge holder, and decortication was performed. The decortication technique performed via VATS is the same as that used in OT. We use the same instruments as in the operating room, including a sponge holder, peanut dissector sponge, and periosteum dissector, as shown in Figs 2 and 3. VATS instruments may be used in deep areas where conventional instruments cannot adequately grasp the tissue, as shown in Fig 3. In some cases, necrotic lung parenchyma was excised if necessary. Throughout the procedure, the lung was completely collapsed to provide optimal exposure. However, intermittent inflation with low tidal volume ventilation was performed when needed to facilitate precise dissection and to prevent iatrogenic lung injury. Blunt dissection was then carried out, followed by flushing the thoracic cavity with warm saline. After the operation was completed, the lung was inflated, and air leaks and bleeding were checked. Upon completion of the procedure and stabilization of hemodynamics, two chest tubes were placed through the utility port and camera port to drain fluid and air leaks. The thoracic wall was then closed in layers, an aseptic dressing was applied, and extubation was promptly performed, if possible, depending on the patient's condition as assessed by the anesthesiologist.

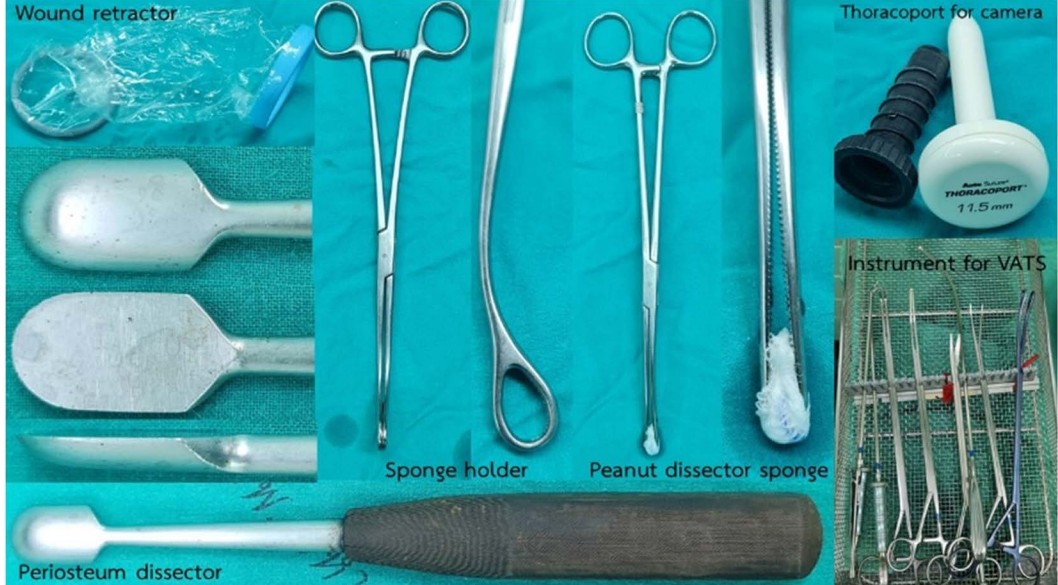

**Fig 2. Instruments used for decortication.**

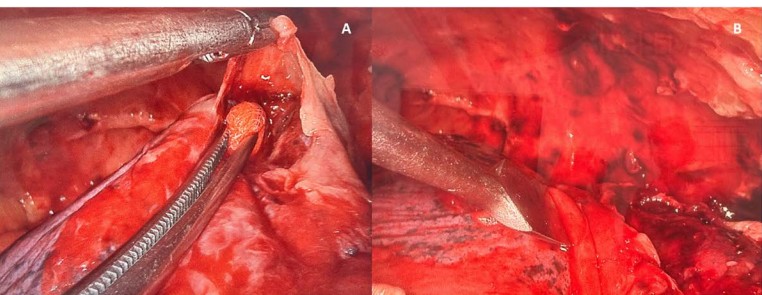

**Fig 3. Decortication performed using a peanut dissector sponge (A) and periosteum dissector (B).**

## Surgical technique of open thoracotomy

Patients received general anesthesia with double-lumen endotracheal intubation if lung isolation was needed. The patient was positioned in the contralateral decubitus position to provide optimal access to the thoracic cavity. A posterolateral or lateral thoracotomy incision was made, and a rib spreader was used to gain greater access to the thoracic cavity. The serratus anterior was divided but latissimus dorsi muscles was spared. Decortication was performed using a sponge holder, peanut dissector sponge, and periosteum dissector, following the same technique as in the VATS approach. The pleural space was irrigated with warm normal saline. After the operation was completed, the lung was inflated, and air leaks and bleeding were checked. Two No. 28 intercostal drainage tubes were inserted in appropriate positions before closing the chest wall. The thoracic wall was then closed in layers, an aseptic dressing was applied, and extubation was promptly performed, if possible, depending on the patient's condition as assessed by the anesthesiologist.

## Postoperative anesthetic management

Our institutional protocol emphasizes early extubation for both VATS and OT approaches, with the goal of extubating patients in the operating room following confirmation of stable respiratory function and hemodynamic parameters. Patients are not routinely maintained on mechanical ventilation postoperatively, except in cases of inadequate ventilation, persistent hemodynamic instability, or other perioperative complications. Early extubation is implemented to minimize the risks associated with prolonged intubation. Following surgery, patients are transferred to the recovery unit and provided with supplemental oxygen and continuous monitoring. Postoperative analgesia is achieved primarily with intravenous opioids, supplemented by intercostal nerve blocks when necessary. Chest radiographs and clinical evaluations are performed to assess lung re-expansion. If satisfactory oxygenation and lung inflation are achieved, re-intubation is not indicated and standard postoperative care is continued.

## Outcome definition

The primary outcomes were the differences between VATS and OT in perioperative outcomes, including operative time, blood loss, perioperative blood transfusion, perioperative complications, and lung expansion. Postoperative outcomes included the length of ICU stay, ventilator dependence, postoperative complications, postoperative blood transfusion, postoperative pain score, chest drain duration, hospital stay, discharge day, and follow-up chest X-rays. Pain scores were assessed using the Visual Analog Scale by a team of well-trained nursing staff who were not involved in the study. Assessments were conducted at the immediate postoperative period, as well as at 8, 24, and 48 hours postoperatively, and at the time of discharge. Complete decortication was defined as the meticulous removal of the organized fibrous peel from visceral pleural surfaces to achieve full lung re-expansion [6]. The extent of decortication was confirmed intraoperatively by the operating surgeon through direct inspection and assessment of pulmonary re-expansion. This evaluation was based on intraoperative findings and postoperative chest radiographs demonstrating full lung re-expansion, as no universally accepted standardized scoring system was applied across all cases. Lung expansion was assessed primarily through postoperative chest radiographs. Full expansion was defined as the absence of residual pleural space or significant pneumothorax, with restoration of the lung contour to the chest wall. Nearly full expansion referred to cases with minimal residual pleural space without significant collapse or evidence of trapped lung. These assessments were made by the operating surgeon together with the attending radiologist.

## Statistical analysis

Patients were divided into two groups: VATS and OT. Categorical data were presented as frequencies and percentages, with comparisons performed using Fisher's exact test. Continuous data were presented as means and standard deviations or medians and interquartile ranges (IQR), depending on the data distribution. Comparisons of continuous variables were performed using either the Student's t-test or the Wilcoxon rank-sum test.

To account for potential confounding factors affecting the choice of surgical method (VATS vs. OT), propensity score matching was employed. Propensity scores were calculated using logistic regression, considering variables such as age, gender, smoking status, Charlson Comorbidity Index (CCI), duration of symptoms, multi-loculation from CT scan, number of loculations, thick visceral pleural peel, preoperative respiratory failure, and lung parenchymal disease. One-to-one propensity score matching was then performed. A standardized mean difference (SMD) between groups for all covariates was calculated to assess balance. A multicollinearity test was also conducted. Statistical significance was determined with a p-value of <0.05. All statistical analyses were performed using STATA version 16.0 (StataCorp, College Station, TX, USA).

## Results

Twenty patients were excluded from this study due to conversion from VATS to OT, resulting in a conversion rate of 4.8% (20 out of 214 patients). The reason for conversion was the presence of a very thick, organized peel covering the lung parenchyma, preventing nearly complete or complete decortication in the non-expanded lung under 30 mmHg positive pressure ventilation. Consequently, 393 patients were included in this study, with 199 in the VATS group and 194 in the OT group.

Table 1 provides a comparison of patient characteristics between the two groups. For the all-patient cohort, there was no significant difference in terms of previous thoracic surgery, laterality of empyema, clinical presentation, cause of thoracic empyema (pneumonia), pre-operative respiratory failure, and underlying diseases except for chronic liver disease, chronic kidney disease, and history of solid tumors. However, the VATS group comprised more elderly and female patients, exhibited a higher Charlson comorbidity index, included a greater proportion of non-smokers, experienced shorter durations of symptoms before diagnosis, had more cases of thick visceral pleural peel, and had a higher prevalence of unidentified causative organisms. Regarding to preoperative CT scans, the proportion of patients who underwent preoperative CT scanning in the VATS group was higher than that in OT group. There was a significant difference in propensity scores between the two groups. After matching for propensity scores (PS), the cohort consisted of 106 patients in each group, with patient characteristics evenly distributed. The propensity score was not statistically different between the two groups post-matching.

The perioperative and postoperative outcomes compared between the two groups are shown in Table 2. Across all patients, no significant differences were observed in terms of complete decortication, intraoperative blood loss, postoperative complications (including air leak, atelectasis, wound infection, and septic shock), perioperative blood transfusions, postoperative pain scores (Fig 4), length of ICU stays, immediate post-surgery lung expansion from chest X-ray, duration of chest drain placement, and lung expansion two weeks after discharge. No in-hospital mortality was observed in either group. However, the VATS group experienced longer operative times, a higher incidence of postoperative pneumonia, a lower rate of re-operation due to clotted hemothorax, increased perioperative platelet transfusions, lower immediate postoperative pain scores, a higher rate of full lung expansion after discharge (as evaluated by chest X-ray), and longer hospital stays. In the propensity-matched cohort analysis, there were no significant differences in most parameters between the groups except for operative time, length of ICU stays, and the rate of full lung expansion at discharge. The operative time in the VATS group was longer than in the OT group; however, the length of ICU stays was shorter, and the rate of full lung expansion at discharge was higher in the VATS group.

Table 3 presents the results of multivariable regression analysis comparing outcome variables between the two groups. For primary analysis, across all patients, there were no notable differences between the groups regarding intraoperative blood loss, duration of ICU stays, operative times, postoperative reduction in pain scores, and full lung expansion at two weeks thereafter. However, the VATS group exhibited significantly full lung expansion at discharge comparing to OT group. In the propensity-matched cohort, there were no significant disparities in intraoperative blood transfusion, postoperative pain reduction, operative time, duration of ICU stays, or full lung expansion at two weeks post-discharge.

**Table 1. Patient characteristics between two groups (N = 393).**

| Variables | All patient cohort | | | | Propensity-matched cohort | | | |
|---|---|---|---|---|---|---|---|---|
| | VATS (n = 199) | OT (n = 194) | P-value | SMD | VATS (n = 106) | OT (n = 106) | P-value | SMD |
| Age (years), mean (SD) | 55.12 ± 15.47 | 47.77 ± 19.48 | <0.001 | 0.418 | 52.31 ± 15.65 | 50.92 ± 19.77 | 0.570 | 0.078 |
| Gender, n (%) | | | <0.001 | 0.451 | | | 0.873 | 0.044 |
| Male | 133 (66.83) | 166 (85.57) | | | 79 (74.53) | 81 (76.42) | | |
| Female | 66 (33.17) | 28 (14.43) | | | 27 (25.47) | 25 (23.58) | | |
| Charlson comorbidity index, Median (IQR) | 2 (1-4) | 1 (0-3) | <0.001 | 0.486 | 2 (0-4) | 2 (1-3) | 0.856 | 0.083 |
| Underlying disease, n (%) | | | | | | | | |
| No underlying disease | 141 (70.85) | 132 (68.04) | 0.585 | 0.061 | 75 (70.75) | 67 (63.21) | 0.307 | 0.161 |
| Coronary heart disease | 9 (4.52) | 6 (3.09) | 0.601 | 0.075 | 3 (2.83) | 5 (4.72) | 0.721 | 0.099 |
| Congestive heart failure | 6 (3.02) | 3 (1.55) | 0.503 | 0.098 | 4 (3.77) | 2 (1.89) | 0.683 | 0.114 |
| Cerebrovascular disease | 11 (5.53) | 5 (2.58) | 0.201 | 0.150 | 3 (2.83) | 4 (3.77) | 1.000 | 0.053 |
| COPD | 3 (1.51) | 3 (1.55) | 1.000 | 0.003 | 1 (0.94) | 2 (1.89) | 1.000 | 0.080 |
| Chronic liver disease | 15 (7.54) | 5 (2.58) | 0.037 | 0.227 | 5 (4.72) | 5 (4.72) | 1.000 | <0.001 |
| Diabetic mellitus | 35 (17.59) | 25 (12.89) | 0.209 | 0.131 | 19 (17.92) | 14 (13.21) | 0.449 | 0.130 |
| Chronic kidney disease | 19 (9.55) | 7 (3.61) | 0.024 | 0.241 | 7 (6.60) | 7 (6.60) | 1.000 | <0.001 |
| History of solid tumor | 32 (16.08) | 15 (7.73) | 0.013 | 0.260 | 14 (13.21) | 11 (10.38) | 0.671 | 0.088 |
| History of pulmonary tuberculosis | 4 (2.01) | 5 (2.58) | 0.748 | 0.038 | 3 (2.83) | 1 (0.94) | 0.621 | 0.139 |
| Previous thoracic surgery, n (%) | 19 (9.55) | 16 (8.25) | 0.724 | 0.046 | 9 (8.49) | 7 (6.60) | 0.796 | 0.071 |
| Smoking status, n (%) | | | 0.045 | 0.276 | | | 0.990 | 0.043 |
| Non-smoker | 125 (62.81) | 98 (50.52) | | | 57 (53.77) | 59 (55.66) | | |
| Ex-smoker | 49 (24.62) | 59 (30.41) | | | 33 (31.13) | 31 (29.25) | | |
| Active smoker | 23 (11.56) | 36 (18.56) | | | 15 (14.15) | 15 (14.15) | | |
| Passive smoker | 2 (1.01) | 1 (0.52) | | | 1 (0.94) | 1 (0.94) | | |
| Laterality of empyema, n (%) | | | 1.000 | 0.005 | | | 0.897 | 0.077 |
| Right | 105 (52.76) | 102 (52.58) | | | 57 (53.77) | 53 (50.00) | | |
| Left | 89 (44.72) | 87 (44.85) | | | 46 (43.40) | 50 (47.17) | | |
| Bilateral | 5 (2.51) | 5 (2.58) | | | 3 (2.83) | 3 (2.83) | | |
| Duration of symptom prior to diagnosis (Days), median (IQR) | 7 (3-14) | 14 (7-30) | <0.001 | 0.330 | 7 (3-30) | 12 (7-30) | 0.049 | 0.077 |
| Presentation, n (%) | | | | | | | | |
| Fever | 98 (49.25) | 112 (57.73) | 0.106 | 0.171 | 56 (52.83) | 62 (58.49) | 0.490 | 0.114 |
| Cough | 87 (43.72) | 97 (50.00) | 0.226 | 0.126 | 48 (45.28) | 49 (46.23) | 1.000 | 0.019 |
| Dyspnea | 146 (73.37) | 128 (65.98) | 0.125 | 0.161 | 77 (72.64) | 73 (68.87) | 0.651 | 0.083 |
| Chest pain | 76 (38.19) | 77 (39.69) | 0.836 | 0.031 | 43 (40.57) | 38 (35.85) | 0.572 | 0.097 |
| Causative organisms, n (%) | | | 0.001 | 0.471 | | | 0.018 | 0.537 |
| Non-identified (no growth) | 134 (67.34) | 116 (59.79) | | | 70 (66.04) | 62 (58.49) | | |
| Tuberculosis | 1 (0.50) | 3 (1.55) | | | 1 (0.94) | 0 | | |
| Staphylococcus | 6 (3.02) | 11 (5.67) | | | 1 (0.94) | 5 (4.72) | | |
| Streptococcus | 23 (11.56) | 25 (12.89) | | | 18 (16.98) | 18 (16.98) | | |
| Gram-negative | 13 (6.53) | 28 (14.43) | | | 5 (4.72) | 15 (14.15) | | |
| Anaerobic | 3 (1.51) | 7 (3.61) | | | 2 (1.89) | 4 (3.77) | | |
| Others | 19 (9.55) | 4 (2.06) | | | 9 (8.49) | 2 (1.89) | | |

*(Continued)*

**Table 1.** (Continued)

| Variables | All patient cohort | | | | Propensity-matched cohort | | | |
|---|---|---|---|---|---|---|---|---|
| | VATS (n = 199) | OT (n = 194) | P-value | SMD | VATS (n = 106) | OT (n = 106) | P-value | SMD |
| Cause of thoracic empyema, n (%) | | | 0.092 | 0.289 | | | 0.906 | 0.143 |
| Pneumonia | 132 (66.33) | 144 (74.23) | | | 73 (68.87) | 76 (71.70) | | |
| Lung abscess | 22 (11.06) | 15 (7.73) | | | 11 (10.38) | 8 (7.55) | | |
| Infected clotted hemothorax after blunt chest trauma | 9 (4.52) | 13 (6.70) | | | 7 (6.60) | 8 (7.55) | | |
| Esophageal perforation | 9 (4.52) | 2 (1.03) | | | 1 (0.94) | 2 (1.89) | | |
| Others | 27 (13.57) | 20 (10.31) | | | 14 (13.21) | 12 (11.32) | | |
| Pre-operative CT scan finding, n (%) | | | <0.001 | 0.845 | | | 0.984 | 0.045 |
| Not performing CT scan | 54 (27.14) | 128 (65.98) | | | 30 (28.30) | 30 (28.30) | | |
| No multiloculation | 71 (35.68) | 32 (16.49) | | | 31 (29.25) | 33 (31.13) | | |
| Multiloculation | 74 (37.19) | 34 (17.53) | | | 45 (42.45) | 43 (40.57) | | |
| Thick visceral pleural peel | 143 (71.86) | 64 (32.99) | <0.001 | 0.844 | 59 (55.66) | 62 (58.49) | 0.781 | 0.057 |
| Lung parenchymal diseases | 45 (31.25) | 25 (38.46) | 0.343 | 0.152 | 17 (28.33) | 24 (38.71) | 0.254 | 0.119 |
| Bronchopleural fistula | 8 (4.02) | 14 (7.22) | 0.192 | 0.139 | 3 (2.83) | 5 (4.7.) | 0.721 | 0.099 |
| Pre-operative respiratory failure, n (%) | 16 (8.04) | 11 (5.67) | 0.427 | 0.094 | 7 (6.60) | 7 (6.60) | 1.000 | <0.001 |
| Propensity score, Mean±SD | 0.63±0.21 | 0.38±0.23 | <0.001 | 1.164 | 0.52±0.20 | 0.52±0.21 | 0.985 | 0.003 |

VATS; Video-Assisted Thoracoscopic Surgery, OT; Open Thoracotomy, SMD; Standardized Mean Differentiation, COPD; Chronic Obstructive Pulmonary Disease, CT; Computed Tomography, SD; Standard Deviation, IQR; Interquartile Range.

Nevertheless, the VATS group demonstrated a greater likelihood of full lung expansion at discharge. For the intention-to-treat analytic approach, results across all patients were consistent with the primary analysis, with the exception that patients in the VATS group were more likely to experience a reduction in pain scores compared with those in the OT group. However, in the propensity-matched cohort, there was no statistically significant difference in pain scores, although a trend toward greater pain reduction in the VATS group was observed (Table 3). For the as-treated analytic approach, among all patients, the VATS group demonstrated reduced intraoperative blood loss, shorter operative time, and a higher proportion of fully expanded lungs at discharge. In the propensity-matched cohort, differences in intraoperative blood loss and operative time remained significant; however, the difference in the proportion of fully expanded lungs at discharge did not reach statistical significance. Nevertheless, a higher rate of fully expanded lungs at two weeks post-discharge was observed in the VATS group (Table 3).

## Discussion

Based on current scientific evidence comparing the outcomes of VATS and OT in the management of organizing thoracic empyema, VATS has demonstrated high success rates, with a 92.9% achievement in complete decortication and 89.9% full lung expansion. Patients undergoing VATS experience fewer complications, reduced need for repeat intubation, decreased wound infection rates, and shorter ICU stays compared to those undergoing OT. The choice between VATS and OT in treating organizing thoracic empyema depends on various factors, including the clinical condition, extent of the disease, and surgeon expertise [7]. Current medical guidelines increasingly favor VATS due to its minimally invasive nature, which results in less postoperative pain, shorter hospital stays, and quicker recovery times [8]. However, debates

**Table 2. Perioperative and postoperative findings between the two groups.**

| Variables | All patient cohort | | | Propensity-matched cohort | | |
|---|---|---|---|---|---|---|
| | VATS (n = 199) | OT (n = 194) | P-value | VATS (n = 106) | OT (n = 106) | P-value |
| Decortication, n (%) | | | 0.760 | | | 1.000 |
| Incomplete decortication due to massive adhesion | 6 (3.02) | 7 (3.61) | | 3 (2.83) | 3 (2.83) | |
| Nearly complete decortication | 4 (2.01) | 6 (3.09) | | 4 (3.77) | 5 (4.72) | |
| Complete decortication | 189 (94.97) | 181 (93.30) | | 99 (93.40) | 98 (92.45) | |
| Operative time (mins), median (IQR) | 120 (85–150) | 95 (75–125) | <0.001 | 130 (90–165) | 105 (75–135) | 0.003 |
| Blood loss (mL), Median (IQR) | 300 (150–500) | 250 (100–500) | 0.088 | 300 (150–500) | 300 (100–400) | 0.270 |
| Postoperative complications, n (%) | | | | | | |
| No postoperative complication | 166 (83.42) | 160 (82.47) | 0.893 | 84 (79.25) | 87 (82.08) | 0.728 |
| Air leak | 15 (7.54) | 9 (4.64) | 0.293 | 2 (1.89) | 1 (0.94) | 1.000 |
| Re-operation due to clotted-hemothorax | 6 (3.02) | 16 (8.25) | 0.028 | 4 (3.77) | 4 (3.77) | 1.000 |
| Pneumonia | 6 (3.02) | 0 | 0.030 | 3 (2.83) | 0 | 0.246 |
| Atelectasis | 3 (1.51) | 1 (0.52) | 0.623 | 2 (1.89) | 1 (0.94) | 1.000 |
| Wound Infection | 3 (1.51) | 1 (0.52) | 0.623 | 3 (2.83) | 0 | 0.246 |
| Septic shock | 1 (0.50) | 3 (1.55) | 0.367 | 1 (0.94) | 3 (2.83) | 0.621 |
| Perioperative PRC transfusion, n (%) | 117 (58.79) | 104 (53.61) | 0.311 | 56 (52.83) | 58 (54.72) | 0.890 |
| Amount of perioperative PRC transfusion (ml), Median (IQR) | 542.5 (290–865) | 540 (290–800) | 0.756 | 506 (280–865) | 565 (360–1060) | 0.319 |
| Perioperative FFP transfusion, n (%) | 32 (16.08) | 37 (19.07) | 0.508 | 18 (16.98) | 22 (20.75) | 0.599 |
| Amount of FFP perioperative transfusion (ml), Median (IQR) | 826 (280–1946) | 645.5 (419.5–1097.5) | 0.657 | 548 (227–980) | 865 (585–1145) | 0.186 |
| Perioperative platelet transfusion, n (%) | 10 (5.03) | 2 (1.03) | 0.036 | 6 (5.66) | 2 (1.89) | 0.280 |
| Amount of perioperative platelet transfusion (ml), Median (IQR) | 280 (160–360) | 135 (135−135) | 0.272 | 317.5 (217.5–517.5) | 135 (135−135) | 0.157 |
| Postoperative pain score, median (IQR) | | | | | | |
| Immediate post-operation | 3 (0–5) | 4 (0–5) | 0.010 | 3 (0–5) | 3 (0–5) | 0.457 |
| Post-operative 8 hrs | 3 (0–5) | 2.5 (0–5) | 0.654 | 3 (0–5) | 2 (0–4) | 0.126 |
| Post-operative 24 hrs | 2 (0–4) | 2 (0–4) | 0.382 | 2 (0–4) | 2 (0–3) | 0.225 |
| Post-operative 48 hrs | 2 (0–3) | 2 (0–4) | 0.165 | 2 (0–3) | 2 (0–4) | 0.671 |
| Pain score at discharge | 0 (0−0) | 0 (0−0) | 0.009 | 0 (0−0) | 0 (0−0) | 0.001 |
| ICU stay (day), median (IQR) | 1 (0–4) | 1 (0–4) | 0.640 | 0 (0–3) | 1 (0–5) | 0.008 |
| Immediate postoperative lung expansion from Chest x-ray, n (%) | | | 0.897 | | | 0.267 |
| Partially expanded | 36 (18.09) | 37 (19.7) | | 14 (13.21) | 21 (19.81) | |
| Complete expanded | 163 (81.91) | 157 (80.93) | | 92 (86.79) | 85 (80.19 | |
| Chest drain duration (days), Median (IQR) | 6 (4–10) | 6 (4–9) | 0.428 | 6 (4–9) | 6 (5–9) | 0.766 |
| Lung expansion at discharge date from Chest x-ray, n (%) | | | <0.001 | | | 0.002 |
| Fully expanded | 179 (89.95) | 136 (70.10) | | 95 (89.62) | 76 (71.70) | |
| Nearly full expanded | 20 (10.05) | 58 (29.90) | | 11 (10.38) | 30 (28.30) | |
| Length of hospital stay (days), Median (IQR) | 20 (11–35) | 14 (10–25) | <0.001 | 19 (11–28) | 16 (11–26) | 0.415 |
| Lung expansion at 2-weeks after discharge from Chest x-ray, n (%) | | | 0.123 | | | 1.000 |
| Fully expanded | 180 (90.45) | 165 (85.05) | | 92 (86.79) | 91 (85.85) | |
| Nearly full expanded | 19 (9.55) | 29 (14.95) | | 14 (13.21) | 15 (14.15) | |

VATS; Video-Assisted Thoracoscopic Surgery, OT; Open Thoracotomy, SMD; Standardized Mean Differentiation, PRC; Pack red cell, FFP; Fresh Frozen Plasma, SD; Standard Deviation, IQR; Interquartile Range,

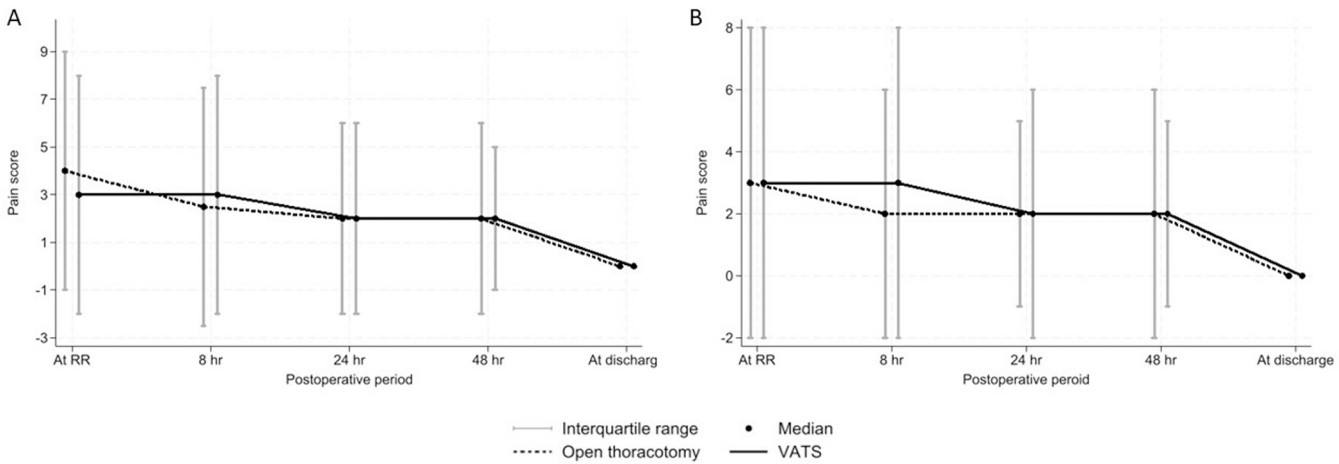

**Fig 4. Postoperative pain score compared between two approaches.** (A) full patient cohort and (B) Propensity-matched patient cohort.

persist regarding the safety and efficacy of VATS, particularly in stage III cases where there is a risk of conversion back to OT due to unsuccessful VATS procedures. While some studies support OT as the gold standard for stage III cases [9], others argue that conversion to OT should be viewed as a safety measure rather than a failure of VATS [8]. Incomplete decortication causing non-expanded lung during VATS indicates the need for conversion to OT, which introduces bias by indication. Therefore, we excluded these patients for our primary analysis to explore the efficacy of VATS for decortication in stage III disease. However, to better reflect real-world clinical practice, a sensitivity analysis including conversion cases was performed using both intention-to-treat and as-treated analytic approaches. The results of these analyses continued to demonstrate the benefits of VATS over the OT group. Regarding to pre-operative CT scan, due to hospital policy during the study period, preoperative CT scans were not routinely performed for all cases of thoracic empyema. CT imaging was typically reserved for patients deemed to be at higher risk, such as those of advanced age, those with significant comorbidities, or those with suspected underlying pulmonary disease or malignancy. Regardless of preoperative imaging, the surgical objective in all cases was to achieve complete decortication and maximal lung re-expansion. Therefore, the absence of preoperative CT scans is unlikely to have significantly influenced the surgical approach or outcomes. Additionally, while some selection bias may have occurred, propensity score matching was performed, resulting in a balanced proportion of preoperative CT imaging between the groups after matching as shown in Table 1.

The results from our study found that postoperative outcomes, such as the proportion of patients undergoing complete decortication and the incidence of perioperative blood transfusion, showed no statistically significant differences between the VATS and OT groups in both the all-patient cohort and the propensity-matched cohort. Multiple variable regression analysis revealed that VATS was associated increased the likelihood of achieving fully expanded lung at discharge in the propensity-matched cohort, indicating favorable respiratory outcomes. Furthermore, in the as-treated analysis, VATS was associated with reduced intraoperative blood loss, shorter operative time, and a higher proportion of fully expanded lungs at two weeks post-discharge. Overall, despite baseline disparities, specific postoperative outcomes were comparable between the two surgical approaches, emphasizing the potential advantages of VATS in certain aspects of managing organizing thoracic empyema.

Comparing these results with other studies, the findings align with other evidence suggesting that VATS may offer benefits in terms of postoperative recovery and respiratory outcomes. For instance, a meta-analysis found that VATS for thoracic empyema was associated with shorter hospital stays and lower complication rates compared to OT [10]. Similarly, another study highlighted the advantages of VATS in terms of reduced postoperative pain and faster recovery times [8].

**Table 3. Multivariable regression analysis for outcome variables comparing between the two groups (VATS versus OT).**

| Outcome variables | All patient cohort | | | Propensity-matched cohort | | |
|---|---|---|---|---|---|---|
| | Estimate | 95% CI | p-value | Estimate* | 95% CI | p-value |
| **Primary analysis (Per-protocol analytic approach)** | | | | | | |
| Intraoperative blood loss (ml) | −35.58[a] | −141.17, 70.12 | 0.508 | −34.20 | −162.89, 94.49 | 0.601 |
| Operative time (min) | −6.37[a] | −27.17, 14.44 | 0.548 | 1.25 | −24.42, 26.91 | 0.924 |
| ICU stay (day) | 1.38[a] | −0.68, 3.44 | 0.189 | 0.68 | −1.56, 3.23 | 0.494 |
| Decreased pain score | 0.71[b] | −0.04, 1.46 | 0.065 | 0.45 | −0.29, 1.18 | 0.236 |
| Fully expanded lung at discharge date | 1.24[c] | 1.07, 1.45 | 0.006 | 1.21 | 1.01, 1.45 | 0.038 |
| Fully expanded lung at 2 weeks after discharge | 1.08[c] | 0.95, 1.23 | 0.221 | 1.14 | 0.97, 1.34 | 0.117 |
| **Intention-to-treat analytic approach** | | | | | | |
| Intraoperative blood loss (ml) | −3.34[a] | −114.85, 108.16 | 0.953 | −7.42 | −125.53, 110.69 | 0.902 |
| Operative time (min) | −7.47[a] | −28.57, 13.63 | 0.487 | −4.84 | −27.67, 17.99 | 0.677 |
| ICU stay (day) | 1.38[a] | −0.58, 3.34 | 0.167 | 1.87 | −0.23, 3.97 | 0.080 |
| Decreased pain score | 0.84[b] | 0.10, 1.57 | 0.025 | 0.78 | −0.01, 1.56 | 0.051 |
| Fully expanded lung at discharge date | 1.25[c] | 1.07, 1.46 | 0.004 | 1.20 | 1.03, 1.41 | 0.023 |
| Fully expanded lung at 2 weeks after discharge | 1.07[c] | 0.94, 1.21 | 0.309 | 1.10 | 0.98, 1.24 | 0.104 |
| **As treated analytic approach** | | | | | | |
| Intraoperative blood loss (ml) | −179.76[a] | −276.167, −83.36 | <0.001 | −215.29 | −331.08, −99.50 | <0.001 |
| Operative time (min) | −24.85[a] | −43.25, −6.45 | 0.008 | −23.27 | −43.64, −2.91 | 0.025 |
| ICU stay (day) | 1.24[a] | −0.48, 2.96 | 0.157 | 1.39 | −0.44, 3.22 | 0.135 |
| Decreased pain score | 0.22[b] | −0.21, 0.65 | 0.314 | 0.71 | −0.09, 1.50 | 0.083 |
| Fully expanded lung at discharge date | 1.19[c] | 1.03, 1.38 | 0.016 | 1.20 | 0.99, 1.45 | 0.062 |
| Fully expanded lung at 2 weeks after discharge | 1.07[c] | 0.95, 1.21 | 0.254 | 1.30 | 1.11, 1.53 | 0.002 |

Intention-to-treat analysis: Patients who underwent conversion from VATS to open thoracotomy were included in the VATS group.

As-treated analysis: Patients who were converted from VATS to open thoracotomy were included in the open thoracotomy group.

[a] Analyzed by multiple linear regression analysis adjusted by multi-loculation, thick pleural peel, year of surgery, and duration symptoms prior to diagnosis, and presented with mean difference (MD) with 95% confidence intervals.

[b] Analyzed by repeated measurements method adjusted by age, gender, operative time, year of surgery, and duration symptoms prior to diagnosis.

[c] Analyzed by exponential risk regression analysis adjusted by multi-loculation, thick pleural peel, year of surgery, and duration symptoms prior to diagnosis, and presented with risk ratio (RR) with 95% confidence intervals.

* Adjusted for year of surgery and duration symptoms prior to diagnosis.

However, a notable distinction observed in this study, compared to certain previous reports, is the longer operative time associated with VATS in the propensity-matched cohort, although this difference did not reach statistical significance. While the VATS approach initially appeared to require extended operative time for decortication compared to OT, increased surgeon experience and progression along the learning curve contributed to improved efficiency over time. After adjustment for operative period in our analysis, no significant difference in operative time was observed between the groups in either the full cohort or the propensity-matched cohort in the primary analysis. This finding contrasts with previous studies that have reported shorter operative times with the VATS approach [11]. In the as-treated analytic approach, VATS was associated with shorter operative times; however, it is important to note that conversion cases-often representing more complex cases or those in which complete decortication could not be achieved via VATS approach-were included in the OT group, potentially contributing to the longer operative times observed in this group. The variation in operative time and other outcomes may result from differences in surgical techniques, patient populations, the thickness of organized peels covering lung, or institutional practices of VATS procedures [12]. A significant concern is the substantial

learning curve associated with the minimally invasive approach, posing a high risk of conversion to thoracotomy, with rates reaching up to 86% [13]. Moreover, VATS has limitations in the management of organizing thoracic empyema due to challenges in accessing the thoracic cavity caused by strong adhesions between the organized peels covering the parietal pleura and the visceral pleura. This difficulty can impede achieving sufficient pleural decortication necessary for lung re-expansion [14]. However, in this study show high successful rate of complete decortication in VATs both the entire patient cohort (94.97%, P = 0.760) and the propensity-matched cohort (93.40%, P = 1.00), although this was not statistically significant. Nevertheless, the possibility of better results after surgery, such as shorter stays in the intensive care unit (ICU) and improved lung expansion upon discharge, highlights the benefits of using VATS for treating organizing thoracic empyema in this study.

The study is strengthened by the analysis of various outcomes, including success rates in complete decortication, lung expansion, ICU stay duration, and perioperative complications. The inclusion of both an entire patient cohort and a propensity-matched cohort enhances the reliability of the comparisons by attempting to control for baseline differences between the VATS and OT groups. Additionally, the study's findings align with existing literature, supporting the growing preference for VATS in clinical guidelines due to its minimally invasive nature and potential benefits in postoperative recovery and respiratory outcomes.

However, this study has several limitations. The retrospective design and non-randomized allocation may introduce bias and limit causal conclusions. Information bias from reliance on medical records is possible. Despite a moderate sample size, the study may lack power for detecting smaller differences, especially in the propensity-matched cohort analysis. Variability in operative time suggests institutional practices and surgeon expertise influence outcomes, emphasizing the learning curve associated with VATS. While attempts to control baseline differences through propensity matching were made, significant demographic and clinical variations could impact outcomes. The transition from open surgery to VATS is associated with a well-recognized learning curve, and varying levels of surgeon experience may influence surgical outcomes, particularly during the early phase following the adoption of VATS. In addition, perioperative care protocols may have evolved during the study period, possibly affecting patient outcomes. We attempted to minimize these confounding effects by controlling for the year of surgery in statistical analyses. The findings of no significant statistical difference in certain outcomes indicate that VATS and OT have comparable effectiveness and safety profiles in certain aspects.

Another limitation is the subjective assessment of lung expansion. The criteria for "full" and "nearly full" expansion were based on the operating surgeon's and attending radiologist's evaluation of postoperative chest radiographs. No formal inter-observer variability analysis was conducted, which may introduce some degree of observer bias in the categorization of lung expansion. Additionally, despite propensity score matching, patients in the OT group had a significantly longer duration of symptoms, suggesting more advanced disease at the time of surgery. This difference could have influenced clinical outcomes and may, in part, account for the comparable results observed between the OT and VATS groups. Therefore, residual confounding due to differences in baseline disease severity cannot be completely excluded, even though this variable was adjusted for in the multivariable analysis. Data on recurrence, mortality, and readmission were not comprehensively available for all patients. This limitation may affect the completeness of outcome assessment in our study.

Despite these limitations, the findings suggest that VATS and OT offer comparable effectiveness and safety in selected outcomes. Notably, VATS was associated with greater lung expansion at discharge, supporting its use for appropriate patients. Importantly, initial VATS can be attempted safely, with conversion to OT as a viable option if necessary.

While VATS offers benefits, OT remains essential in select cases. It is preferred for advanced-stage disease, dense or calcified pleural peel, extensive adhesions, or when visualization is limited. OT also allows more thorough debridement and better management in complex or re-operative cases. Thus, it remains a vital option for appropriately selected patients with thoracic empyema.

Although propensity score matching and multivariable adjustment were used in this study to minimize confounding, the possibility of residual confounding from unmeasured variables still exists. This limitation should be considered when interpreting the results of this study.

## Conclusion

In summary, this study suggests that VATS demonstrates advantages over open thoracotomy in the management of organizing thoracic empyema, including shorter postoperative ICU stays and a higher likelihood of lung expansion at discharge, although VATS is associated with longer operative times. However, the overall effectiveness and safety profiles of VATS and OT appear similar, as evidenced by comparable rates of complete decortication, perioperative blood transfusion needs, pain scores, intraoperative blood loss, and lung expansion at two weeks, particularly when comparing well-matched groups of patients. These findings support the consideration of VATS as a viable surgical option in appropriate patients with organizing thoracic empyema.

While this study contributes valuable insights, it is essential to acknowledge its limitations, including the relatively moderate sample size and retrospective nature. Although propensity score analysis and multivariable adjustment were utilized to address confounding factors, certain potential sources of bias inherent to retrospective studies may remain. To strengthen the evidence base and provide more robust conclusions, further studies with larger sample sizes or randomized controlled trials are warranted to validate and expand upon these findings. Such investigations would enhance our understanding of the comparative effectiveness of VATS and OT in treating organizing thoracic empyema.

## Supporting information

**S1 File. Minimal data set underlying the findings described in the manuscript.**
(XLSX)

## Acknowledgments

During the preparation of this work the author(s) used ChatGPT in order to improve medical writing English and grammar error. After using this tool, the author(s) reviewed and edited the content as needed and take full responsibility for the content of the publication.

## Author contributions

**Conceptualization:** Thanat Suwanarawat, Rachata Deesang, Sasikarn Tawanseree, Sikkawat Watcharasoonthonkit, Somcharoen Saeteng, Sophon Siwachat, Apichat Tantraworasin.

**Data curation:** Thanat Suwanarawat, Rachata Deesang, Sasikarn Tawanseree, Sikkawat Watcharasoonthonkit, Somcharoen Saeteng, Sophon Siwachat, Apichat Tantraworasin.

**Formal analysis:** Thanat Suwanarawat, Rachata Deesang, Sasikarn Tawanseree, Sikkawat Watcharasoonthonkit, Apichat Tantraworasin.

**Investigation:** Apichat Tantraworasin.

**Methodology:** Thanat Suwanarawat, Rachata Deesang, Sasikarn Tawanseree, Sikkawat Watcharasoonthonkit, Somcharoen Saeteng, Sophon Siwachat, Apichat Tantraworasin.

**Project administration:** Apichat Tantraworasin.

**Resources:** Somcharoen Saeteng, Sophon Siwachat, Apichat Tantraworasin.

**Software:** Apichat Tantraworasin.

**Supervision:** Somcharoen Saeteng, Apichat Tantraworasin.

**Validation:** Thanat Suwanarawat, Rachata Deesang, Sasikarn Tawanseree, Sikkawat Watcharasoonthonkit, Somcharoen Saeteng, Sophon Siwachat, Apichat Tantraworasin.

**Visualization:** Thanat Suwanarawat, Rachata Deesang, Sasikarn Tawanseree, Sikkawat Watcharasoonthonkit, Somcharoen Saeteng, Sophon Siwachat, Apichat Tantraworasin.

**Writing – original draft:** Thanat Suwanarawat, Rachata Deesang, Sasikarn Tawanseree, Sikkawat Watcharasoonthonkit, Sophon Siwachat, Apichat Tantraworasin.

**Writing – review & editing:** Thanat Suwanarawat, Rachata Deesang, Sasikarn Tawanseree, Sikkawat Watcharasoonthonkit, Somcharoen Saeteng, Sophon Siwachat, Apichat Tantraworasin.

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
