## [Decision Letter · Decision Letter 0]

28 Jul 2025

PONE-D-25-37489
Comparative Analysis of Surgical Outcomes: Video-Assisted Thoracoscopic Surgery versus Open Thoracotomy in Organizing Empyema Thoracis Management
PLOS ONE

Dear Dr. Tantraworasin,

Thank you for submitting your manuscript to PLOS ONE. After careful consideration, we feel that it has merit but does not fully meet PLOS ONE’s publication criteria as it currently stands. Therefore, we invite you to submit a revised version of the manuscript that addresses the points raised during the review process.

We look forward to receiving your revised manuscript.

Kind regards,

Luca Bertolaccini, M.D., Ph.D.

Academic Editor

PLOS ONE

Journal Requirements:

“This study was partially supported by Chiang Mai University, Chiang Mai, Thailand.”

“This study was partially supported by Chiang Mai University, Chiang Mai, Thailand.”

“This study was partially supported by Chiang Mai University, Chiang Mai, Thailand.”

“The author(s) received no specific funding for this work”

6. We note that there is identifying data in the Supporting Information file <DATA_file.xlsx>. Due to the inclusion of these potentially identifying data, we have removed this file from your file inventory. Prior to sharing human research participant data, authors should consult with an ethics committee to ensure data are shared in accordance with participant consent and all applicable local laws.

-Location data

Please remove or anonymize all personal information (<Age>), ensure that the data shared are in accordance with participant consent, and re-upload a fully anonymized data set. Please note that spreadsheet columns with personal information must be removed and not hidden as all hidden columns will appear in the published file.

**Additional Editor Comments:**

The reviewers have emphasised issues that require a careful and thorough manuscript revision.

No commitment to publication can be made at this point.

Reviewers' comments:

Reviewer's Responses to Questions

**Comments to the Author**

1. Is the manuscript technically sound, and do the data support the conclusions?

Reviewer #1: Yes

Reviewer #2: Yes

2. Has the statistical analysis been performed appropriately and rigorously? 

Reviewer #1: Yes

Reviewer #2: Yes

3. Have the authors made all data underlying the findings in their manuscript fully available?

Reviewer #1: Yes

Reviewer #2: Yes

4. Is the manuscript presented in an intelligible fashion and written in standard English?

Reviewer #1: Yes

Reviewer #2: Yes

5. Review Comments to the Author

Reviewer #1: Dear Authors,

Thank you for submitting your manuscript for review. I was pleased to receive it.

Your large dataset and the inclusion of useful perioperative outcomes offer real-world insight into the comparison of VATS and open thoracotomy for organizing empyema. However, I have a few comments:

1. You mention that VATS was introduced in 2015 and became the preferred technique thereafter. Could you comment on whether this shift over time may have introduced a performance/experience bias (e.g., learning curve, evolving perioperative care), and how this may have impacted the internal validity of your findings?

2. Surgeon experience may influence operative time and outcomes, especially given the reported differences in timing of VATS adoption. Were the same surgeons performing both techniques? Was surgical expertise considered in the analysis?

3. What tool did you use for pain measurement (e.g., numeric rating scale, visual analogue scale). Please clarify how pain was assessed, who collected the data, and whether assessors were blinded to the surgical approach.

4. A large proportion of patients in the open thoracotomy group did not undergo preoperative CT scanning. Could this missing imaging data may have influenced surgical planning?

5. Could you explain why you excluded patients who were converted from VATS to open thoracotomy, since conversions are part of real-world VATS outcomes?

6. You state that five patients were excluded due to conversion from VATS to open thoracotomy. Could you also include the overall conversion rate?

7. Could you define “complete decortication”? Was it based on the surgeon’s judgment, or did you apply standardized criteria across all cases?

8. The assessment of lung expansion is central to your conclusions. However, you have not described the criteria used to categorize expansion as “full” or “nearly full”. Could you clarify how you performed these assessments and whether any inter-observer variability was evaluated?

9. I believe you tend to emphasize the benefits of VATS, but open thoracotomy may remain preferable in certain scenarios. Could you discuss potential indications/advantages of open thoracotomy?

10. If data on recurrence, mortality, or readmission are available, consider including them. If not available, consider adding a comment on this limitation.

Thank you again for the opportunity to review your work.

Reviewer #2: I read with great interest your article on the comparative analysis of surgical outcomes between VATS and open thoracotomy for the treatment of thoracic empyema. I would like to congratulate the authors on their results and the large sample size. However, I have several comments and recommendations that may enhance the interest of readers in your work:

· I suggest replacing the term “empyema thoracis” with “thoracic empyema” for improved clarity and consistency with commonly used terminology.

· Line 69: Please correct the syntax of the sentence, as the word “caused” appears twice.

· You state that you only operated patients with stage III empyema. In my experience, it is often difficult to determine the exact stage of a thoracic infection preoperatively, as CT scans, blood tests, or chest drain output may not reliably reflect the presence of pleural thickening. Could you clarify how you identified the appropriate timing for surgery in confirmed stage III cases?

· It would be helpful to describe in greater detail the decortication technique used in both VATS and thoracotomy approaches, including the instruments employed, as this is central to your study.

· Please elaborate on the intraoperative and postoperative anesthetic management of patients. For example:

o Is the lung completely collapsed or slightly inflated to facilitate dissection?

o In some centers, patients remain intubated for 24–48 hours postoperatively to support lung re-expansion. Do you prefer early extubation? Please provide details of your protocol and rationale.

· How do you assess full lung re-expansion, and do you have a defined criterion for complete decortication? These are subjective evaluations, and it is often difficult to assess them reliably via chest X-rays, CT scans, or even intraoperatively. Providing a clear definition would strengthen your methodology.

· Patients who underwent open thoracotomy had a significantly longer duration of symptoms, even after propensity score matching. Could this suggest that similar outcomes between thoracotomy and VATS might be due to more advanced disease in the thoracotomy group? Please address this point in the manuscript.

6. PLOS authors have the option to publish the peer review history of their article (what does this mean?). If published, this will include your full peer review and any attached files.

Reviewer #1: **Yes: **Savvas Lampridis

Reviewer #2: No

---

## [Author Response · Author response to Decision Letter 1]

23 Aug 2025

Reviewer #1:

Comment 1:

You mention that VATS was introduced in 2015 and became the preferred technique thereafter. Could you comment on whether this shift over time may have introduced a performance/experience bias (e.g., learning curve, evolving perioperative care), and how this may have impacted the internal validity of your findings?

Response: We have re-analyzed data and revised results (Table 3), add sentences in discussion section to clarify the research gap as follows:

Table 3. Multivariable regression analysis for outcome variables comparing between the two groups (VATS versus OT)

Outcome variables All patient cohort Propensity-matched cohort

 Estimate 95 % CI p-value Estimate* 95 % CI p-value

Intraoperative blood loss (ml) -35.58a -141.17, 70.12 0.508 -34.20 -162.89, 94.49 0.601

Operative time (min) -6.37a -27.17, 14.44 0.548 1.25 -24.42, 26.91 0.924

ICU stay (day) 1.38a -0.68, 3.44 0.189 0.68 -1.56, 3.23 0.494

Decreased pain score 0.71b -0.04, 1.46 0.065 0.45 -0.29, 1.18 0.236

Fully expanded lung at discharge date 1.24c 1.07, 1.45 0.006 1.21 1.01, 1.45 0.038

Fully expanded lung at 2 weeks after discharge 1.08c 0.95, 1.23 0.221 1.14 0.97, 1.34 0.117

aAnalyzed by multiple linear regression analysis adjusted by multi-loculation, thick pleural peel, year of surgery, and duration symptoms prior to diagnosis, and presented with mean difference (MD) with 95 % confidence intervals.

bAnalyzed by repeated measurements method adjusted by age, gender, operative time, year of surgery, and duration symptoms prior to diagnosis.

cAnalyzed by exponential risk regression analysis adjusted by multi-loculation, thick pleural peel, year of surgery, and duration symptoms prior to diagnosis, and presented with risk ratio (RR) with 95 % confidence intervals.

*Adjusted for year of surgery and duration symptoms prior to diagnosis.

We added sentences in discussion part as shown below;

“However, a notable distinction observed in this study, compared to certain previous reports, is the longer operative time associated with VATS in the propensity-matched cohort, although this difference did not reach statistical significance. While the VATS approach initially appeared to require extended operative time for decortication compared to OT, increased surgeon experience and progression along the learning curve contributed to improved efficiency over time. After adjustment for operative period in our analysis, no significant difference in operative time was observed between the groups in either the full cohort or the propensity-matched cohort. This finding contrasts with studies suggesting shorter operative times with VATS[9]. The variation in operative time and other outcomes may be attributable to differences in surgical techniques, patient populations, the thickness of organized peels covering lung, or institutional VATS practices [10]”

We added sentences in limitation of this study as shown below;

“However, the study has limitations. The retrospective design and non-randomized allocation may introduce bias and limit causal conclusions. Information bias from reliance on medical records is possible. Despite a moderate sample size, the study may lack power for detecting smaller differences, especially in the propensity-matched cohort analysis. The transition from open surgery to VATS is associated with a well-recognized learning curve, and varying levels of surgeon experience may influence surgical outcomes, particularly during the early phase following the adoption of VATS. In addition, perioperative care protocols may have evolved during the study period, possibly affecting patient outcomes. We attempted to minimize these confounding effects by controlling for the year of surgery in statistical analyses.”

Comment 2: Surgeon experience may influence operative time and outcomes, especially given the reported differences in timing of VATS adoption. Were the same surgeons performing both techniques? Was surgical expertise considered in the analysis?

Response: In our study, both VATS and open thoracotomy procedures were performed by a consistent team of thoracic surgeons, all of whom had experience with both techniques. However, the adoption of VATS introduced a learning curve which may have impacted operative times and outcomes, particularly in the initial phase following its implementation. To address the potential influence of surgical expertise, we adjusted for time period of surgery in our analysis as shown in comment 1. Nevertheless, we acknowledge that differences in individual experience and progression along the learning curve are potential sources of bias that could not be fully accounted for in our retrospective analysis. This limitation has been noted in the revised manuscript as shown below;

In patients and methods part, we added sentence “Both VATS and OT procedures were performed by a consistent team of thoracic surgeons, all of whom had experience with both techniques.”

Table 3 which is the multivariable analysis, we adjusted year of surgery in the multivariable model both in full cohort and propensity-matched cohort and add some paragraph in discussion part as shown in comment 1.

Comment 3: What tool did you use for pain measurement (e.g., numeric rating scale, visual analogue scale). Please clarify how pain was assessed, who collected the data, and whether assessors were blinded to the surgical approach.

Response: Postoperative pain was assessed using the Visual Analog Scale (VAS). Pain assessments were performed by a dedicated team of well-trained nursing staff who were not involved in the study. These assessments were routinely conducted at the immediate postoperative period, as well as at 8, 24, and 48 hours after surgery, and at the time of discharge. The nursing staff were not formally blinded to the surgical approach due to the nature of clinical care; however, they were not involved in the study design, data analysis, or patient allocation. We have added these details to the Methods section (Outcome Definition) in the revised manuscript as shown below;

“Pain scores were assessed using the Visual Analog Scale by a team of well-trained nursing staff who were not involved in the study. Assessments were conducted at the immediate postoperative period, as well as at 8, 24, and 48 hours postoperatively, and at the time of discharge.”

Comment 4: A large proportion of patients in the open thoracotomy group did not undergo preoperative CT scanning. Could this missing imaging data may have influenced surgical planning?

Response: Due to hospital policy during the study period, preoperative CT scans were not routinely performed for all cases of thoracic empyema. CT imaging was typically reserved for patients deemed to be at higher risk, such as those of advanced age, those with significant comorbidities, or those with suspected underlying pulmonary disease or malignancy. Regardless of preoperative imaging, the surgical objective in all cases was to achieve complete decortication and maximal lung re-expansion. Therefore, the absence of preoperative CT scans is unlikely to have significantly influenced the surgical approach or outcomes. Additionally, while some selection bias may have occurred, propensity score matching was performed, resulting in a balanced proportion of preoperative CT imaging between the groups after matching.

We added this details in result part and discussion part as shown below;

In result part, “Regarding to preoperative CT scans, the proportion of patients who underwent preoperative CT scanning in the VATS group was higher than that in OT group.”

In discussion part, “Regarding to pre-operative CT scan, due to hospital policy during the study period, preoperative CT scans were not routinely performed for all cases of thoracic empyema. CT imaging was typically reserved for patients deemed to be at higher risk, such as those of advanced age, those with significant comorbidities, or those with suspected underlying pulmonary disease or malignancy. Regardless of preoperative imaging, the surgical objective in all cases was to achieve complete decortication and maximal lung re-expansion. Therefore, the absence of preoperative CT scans is unlikely to have significantly influenced the surgical approach or outcomes. Additionally, while some selection bias may have occurred, propensity score matching was performed, resulting in a balanced proportion of preoperative CT imaging between the groups after matching as shown in Table 1.”

Comment 5: Could you explain why you excluded patients who were converted from VATS to open thoracotomy, since conversions are part of real-world VATS outcomes?

Response: We excluded patients who were converted from VATS to open thoracotomy to ensure a clear comparison between the intended VATS and open thoracotomy groups, and to minimize confounding variables related to intraoperative decision-making and patient selection. We agree that conversions represent important real-world outcomes of VATS; however, our primary aim was to compare perioperative and postoperative outcomes between patients who completed either approach as initially planned. We acknowledge this as a limitation of our study and have now discussed its potential impact in the revised manuscript as follow;

“One notable limitation of our study is the exclusion of patients who were converted from VATS to OT during the procedure. While this approach was intended to facilitate a direct comparison between patients who completed either VATS or OT as originally planned, it may have introduced selection bias. Consequently, our findings may not fully reflect the perioperative and postoperative outcomes associated with conversion cases, which represent important real-world events in VATS practice. Therefore, the generalizability of our results to all patients undergoing VATS, particularly those at higher risk for conversion, may be limited. Future studies should consider including converted cases to provide a more comprehensive assessment of surgical outcomes.”

Comment 6: You state that five patients were excluded due to conversion from VATS to open thoracotomy. Could you also include the overall conversion rate?

Response: The overall conversion rate from VATS to open thoracotomy in our study was 2.5 %, as five out of 204 patients required conversion. We have now included this information in the revised manuscript at result part for clarity as follow;

“Five patients were excluded from this study due to conversion from VATS to OT, resulting in a conversion rate of 2.5 % (5 out of 204 patients).”

Comment 7. Could you define “complete decortication”? Was it based on the surgeon’s judgment, or did you apply standardized criteria across all cases?

Response: In our study, “complete decortication” was defined as the full removal of the fibrous peel from both the visceral and parietal pleura to allow complete lung re-expansion, as confirmed intraoperatively. This assessment was primarily based on the surgeon’s judgment during the procedure. We acknowledge the subjective nature of this definition, as no universally standardized criteria were formally applied across all cases. This clarification has been included in the revised manuscript at the outcome definition part as follow;

“Complete decortication was defined as the meticulous removal of the organized fibrous peel from visceral pleural surfaces to achieve full lung re-expansion. The extent of decortication was confirmed intraoperatively by the operating surgeon through direct inspection and assessment of pulmonary re-expansion. This evaluation was based on intraoperative findings and postoperative chest radiographs demonstrating full lung re-expansion, as no universally accepted standardized scoring system was applied across all cases.”

Comment 8. The assessment of lung expansion is central to your conclusions. However, you have not described the criteria used to categorize expansion as “full” or “nearly full”. Could you clarify how you performed these assessments and whether any inter-observer variability was evaluated?

Response: In our study, lung expansion was assessed primarily through postoperative chest radiographs. "Full expansion" was defined as the absence of residual pleural space or significant pneumothorax, with restoration of the lung contour to the chest wall. "Nearly full expansion" referred to cases with minimal residual pleural space without significant collapse or evidence of trapped lung. These assessments were made by the operating surgeon together with the attending radiologist. However, we acknowledge that no formal inter-observer variability analysis was performed, and the evaluation may be subject to some degree of subjective interpretation. We have clarified these definitions and acknowledged this limitation in the revised manuscript at the outcome definition part as follow;

“Lung expansion was assessed primarily through postoperative chest radiographs. Full expansion was defined as the absence of residual pleural space or significant pneumothorax, with restoration of the lung contour to the chest wall. Nearly full expansion referred to cases with minimal residual pleural space without significant collapse or evidence of trapped lung. These assessments were made by the operating surgeon together with the attending radiologist.”

And I added sentences in the limitation of the study as follow;

“Another limitation is the subjective assessment of lung expansion. The criteria for "full" and "nearly full" expansion were based on the operating surgeon’s and attending radiologist’s evaluation of postoperative chest radiographs. No formal inter-observer variability analysis was conducted, which may introduce some degree of observer bias in the categorization of lung expansion.”

Comment 9. I believe you tend to emphasize the benefits of VATS, but open thoracotomy may remain preferable in certain scenarios. Could you discuss potential indications/advantages of open thoracotomy?

Response: We agree that, despite the recognized benefits of VATS, open thoracotomy retains important roles in the management of pleural empyema and related thoracic conditions. Open thoracotomy may be preferable in cases with advanced-stage disease, dense or calcified pleural peel, extensive adhesions, or when visualization and access are limited during minimally invasive surgery. It also allows for more extensive debridement and better control in challenging anatomical situations or in patients with prior thoracic surgeries. We have now included a discussion of these potential indications and advantages for open thoracotomy in the revised manuscript, to provide a more balanced perspective as follow;

“While VATS offers benefits, OT remains essential in select cases. It is preferred for advanced-stage disease, dense or calcified pleural peel, extensive adhesions, or when visualization is limited. OT also allows more thorough debridement and better management in complex or re-operative cases. Thus, it remains a vital option for appropriately selected patients with thoracic empyema.”

Comment 10. If data on recurrence, mortality, or readmission are available, consider including them. If not available, consider adding a comment on this limitation.

Response: No in-hospital mortality was observed in either group of study. Unfortunately, data on recurrence, mortality, and readmission were not comprehensively available for all patients in our study, primarily due to limitations in retrospective data collection. We have now acknowledged this as a study limitation in the revised manuscript as follow;

“Data on recurrence, mortality, and readmission were not comprehensively available for all patients. This limitation may affect the completeness of outcome assessment in our study.”

I add “in-hospital mortality” in the result part as follow;

“No in-hospital mortality was observed in either group.”

Reviewer #2: I read with great interest your article on the comparative analysis of surgical outcomes between VATS and open thoracotomy for the treatment of thoracic empyema. I would like to congratulate the authors on their results and the large sample size. However, I have several comments and recommendations that may enhance the interest of readers in your work:

Comment 1:

---

## [Decision Letter · Decision Letter 1]

29 Aug 2025

PONE-D-25-37489R1
Comparative Analysis of Surgical Outcomes: Video-Assisted Thoracoscopic Surgery versus Open Thoracotomy in Organizing Thoracic Empyema Management
PLOS ONE

Dear Dr. Tantraworasin,

Thank you for submitting your manuscript to PLOS ONE. After careful consideration, we feel that it has merit but does not fully meet PLOS ONE’s publication criteria as it currently stands. Therefore, we invite you to submit a revised version of the manuscript that addresses the points raised during the review process.

We look forward to receiving your revised manuscript.

Kind regards,

Luca Bertolaccini, M.D., Ph.D.

Academic Editor

PLOS ONE

Journal Requirements:

Reviewer #1:

Thank you for the revisions, which have significantly improved your manuscript. However, there are a few remaining points:

1. I appreciate that you have acknowledged the exclusion of conversion cases, but this remains an important limitation as conversions represent clinically relevant outcomes in real-world practice. A sensitivity analysis including these cases would strengthen the study, but I accept that this may not be feasible with the current dataset.

2. I believe residual confounding is still possible despite propensity matching and multivariable adjustment (e.g., longer duration of symptoms in the thoracotomy group). I suggest highlighting this more clearly in the discussion.

3. There are still typos throughout (e.g., “blood lose” should be “blood loss”).

Reviewer #2:

Reviewer's Responses to Questions

**Comments to the Author**

1. If the authors have adequately addressed your comments raised in a previous round of review and you feel that this manuscript is now acceptable for publication, you may indicate that here to bypass the “Comments to the Author” section, enter your conflict of interest statement in the “Confidential to Editor” section, and submit your "Accept" recommendation.

Reviewer #1: (No Response)

Reviewer #2: All comments have been addressed

2. Is the manuscript technically sound, and do the data support the conclusions?

Reviewer #1: Yes

Reviewer #2: Yes

3. Has the statistical analysis been performed appropriately and rigorously? 

Reviewer #1: Yes

Reviewer #2: Yes

4. Have the authors made all data underlying the findings in their manuscript fully available?

Reviewer #1: Yes

Reviewer #2: Yes

5. Is the manuscript presented in an intelligible fashion and written in standard English?

Reviewer #1: Yes

Reviewer #2: Yes

6. Review Comments to the Author

Reviewer #1: Dear Authors,

Thank you for the revisions, which have significantly improved your manuscript. However, there are a few remaining points:

1. I appreciate that you have acknowledged the exclusion of conversion cases, but this remains an important limitation as conversions represent clinically relevant outcomes in real-world practice. A sensitivity analysis including these cases would strengthen the study, but I accept that this may not be feasible with the current dataset.

2. I believe residual confounding is still possible despite propensity matching and multivariable adjustment (e.g., longer duration of symptoms in the thoracotomy group). I suggest highlighting this more clearly in the discussion.

3. There are still typos throughout (e.g., “blood lose” should be “blood loss”).

Thank you again for your efforts.

Reviewer #2: The authors have adequately revised the manuscript, enhancing its scientific value. The improvements are clear and contribute meaningfully to the clarity and rigor of the work. In my opinion, it can be accepted without further revision

7. PLOS authors have the option to publish the peer review history of their article (what does this mean?). If published, this will include your full peer review and any attached files.

Reviewer #1: **Yes: **Savvas Lampridis

Reviewer #2: No

---

## [Author Response · Author response to Decision Letter 2]

1 Sep 2025

Response to Reviewer Comments

Dear Editors and Reviewers,

We would like to thank you for your careful review and valuable comments on our manuscript, entitled “Comparative Analysis of Surgical Outcomes: Video-Assisted Thoracoscopic Surgery versus Open Thoracotomy in Organizing Thoracic Empyema Management”. We appreciate the time and effort you and the reviewers have dedicated to enhancing the quality of our work.

Please find our responses to each reviewer’s comments below. All changes made in the revised manuscript are highlighted for your convenience.

Reviewer #1:

Thank you for the revisions, which have significantly improved your manuscript. However, there are a few remaining points:

1. I appreciate that you have acknowledged the exclusion of conversion cases, but this remains an important limitation as conversions represent clinically relevant outcomes in real-world practice. A sensitivity analysis including these cases would strengthen the study, but I accept that this may not be feasible with the current dataset.

Response: Thank you for your thoughtful comment and for highlighting the clinical relevance of conversion cases. In line with your suggestion, we have conducted a sensitivity analysis that includes the conversion cases and have incorporated the results into the revised manuscript.

We added the paragraph in the method part as shown below;

“Patients who initially underwent VATS but were converted to OT due to a very thick organized peel covering the visceral pleura, resulting in incomplete decortication, or due to unexpected complications, were excluded from the primary analysis to minimize indication bias (per-protocol approach). However, to reflect real-world practice, a sensitivity analysis that included conversion cases was performed using both intention-to-treat and as-treated analytic approaches.”

We have also corrected the error regarding the conversion cases, which comprised 4.8% of patients, rather than 5 cases as previously stated. In addition, we have revised the Results section and updated Table 3 to reflect this information, as shown below.

“For the intention-to-treat analytic approach, results across all patients were consistent with the primary analysis, with the exception that patients in the VATS group were more likely to experience a reduction in pain scores compared with those in the OT group. However, in the propensity-matched cohort, there was no statistically significant difference in pain scores, although a trend toward greater pain reduction in the VATS group was observed (Table 3). For the as-treated analytic approach, among all patients, the VATS group demonstrated reduced intraoperative blood loss, shorter operative time, and a higher proportion of fully expanded lungs at discharge. In the propensity-matched cohort, differences in intraoperative blood loss and operative time remained significant; however, the difference in the proportion of fully expanded lungs at discharge did not reach statistical significance. Nevertheless, a higher rate of fully expanded lungs at two weeks post-discharge was observed in the VATS group (Table 3).”

“Table 3. Multivariable regression analysis for outcome variables comparing between the two groups (VATS versus OT)

Outcome variables All patient cohort Propensity-matched cohort

 Estimate 95 % CI p-value Estimate* 95 % CI p-value

Primary analysis (Per-protocol analytic approach)

Intraoperative blood loss (ml) -35.58a -141.17, 70.12 0.508 -34.20 -162.89, 94.49 0.601

Operative time (min) -6.37a -27.17, 14.44 0.548 1.25 -24.42, 26.91 0.924

ICU stay (day) 1.38a -0.68, 3.44 0.189 0.68 -1.56, 3.23 0.494

Decreased pain score 0.71b -0.04, 1.46 0.065 0.45 -0.29, 1.18 0.236

Fully expanded lung at discharge date 1.24c 1.07, 1.45 0.006 1.21 1.01, 1.45 0.038

Fully expanded lung at 2 weeks after discharge 1.08c 0.95, 1.23 0.221 1.14 0.97, 1.34 0.117

Intention-to-treat analytic approach

Intraoperative blood loss (ml) -3.34a -114.85, 108.16 0.953 -7.42 -125.53, 110.69 0.902

Operative time (min) -7.47a -28.57, 13.63 0.487 -4.84 -27.67, 17.99 0.677

ICU stay (day) 1.38a -0.58, 3.34 0.167 1.87 -0.23, 3.97 0.080

Decreased pain score 0.84b 0.10, 1.57 0.025 0.78 -0.01, 1.56 0.051

Fully expanded lung at discharge date 1.25c 1.07, 1.46 0.004 1.20 1.03, 1.41 0.023

Fully expanded lung at 2 weeks after discharge 1.07c 0.94, 1.21 0.309 1.10 0.98, 1.24 0.104

As treated analytic approach

Intraoperative blood loss (ml) -179.76a -276.167, -83.36 <0.001 -215.29 -331.08, -99.50 <0.001

Operative time (min) -24.85a -43.25, -6.45 0.008 -23.27 -43.64, -2.91 0.025

ICU stay (day) 1.24a -0.48, 2.96 0.157 1.39 -0.44, 3.22 0.135

Decreased pain score 0.22b -0.21, 0.65 0.314 0.71 -0.09, 1.50 0.083

Fully expanded lung at discharge date 1.19c 1.03, 1.38 0.016 1.20 0.99, 1.45 0.062

Fully expanded lung at 2 weeks after discharge 1.07c 0.95, 1.21 0.254 1.30 1.11, 1.53 0.002

Intention-to-treat analysis: Patients who underwent conversion from VATS to open thoracotomy were included in the VATS group.

As-treated analysis: Patients who were converted from VATS to open thoracotomy were included in the open thoracotomy group.

aAnalyzed by multiple linear regression analysis adjusted by multi-loculation, thick pleural peel, year of surgery, and duration symptoms prior to diagnosis, and presented with mean difference (MD) with 95 % confidence intervals.

bAnalyzed by repeated measurements method adjusted by age, gender, operative time, year of surgery, and duration symptoms prior to diagnosis.

cAnalyzed by exponential risk regression analysis adjusted by multi-loculation, thick pleural peel, year of surgery, and duration symptoms prior to diagnosis, and presented with risk ratio (RR) with 95 % confidence intervals.

*Adjusted for year of surgery and duration symptoms prior to diagnosis.”

We added and delete paragraph in discussion part as shown below.

“However, to better reflect real-world clinical practice, a sensitivity analysis including conversion cases was performed using both intention-to-treat and as-treated analytic approaches. The results of these analyses continued to demonstrate the benefits of VATS over the OT group.”

“Furthermore, in the as-treated analysis, VATS was associated with reduced intraoperative blood loss, shorter operative time, and a higher proportion of fully expanded lungs at two weeks post-discharge.”

“This finding contrasts with previous studies that have reported shorter operative times with the VATS approach [11]. In the as-treated analytic approach, VATS was associated with shorter operative times; however, it is important to note that conversion cases-often representing more complex cases or those in which complete decortication could not be achieved via VATS approach-were included in the OT group, potentially contributing to the longer operative times observed in this group.”

2. I believe residual confounding is still possible despite propensity matching and multivariable adjustment (e.g., longer duration of symptoms in the thoracotomy group). I suggest highlighting this more clearly in the discussion.

Response: Thank you for your insightful comment regarding the possibility of residual confounding, even after propensity matching and multivariable adjustment. We agree with your assessment that unmeasured confounding factors—such as the longer duration of symptoms in the thoracotomy group—may still influence our findings. In response, we have revised the Discussion section to more explicitly acknowledge this limitation. Specifically, we added the following statement:

“Although propensity score matching and multivariable adjustment were used in this study to minimize confounding, the possibility of residual confounding from unmeasured variables still exists. This limitation should be considered when interpreting the results of this study.”

From previous revised manuscript we already discuss this point as shown in the paragraph below:

“Additionally, despite propensity score matching, patients in the OT group had a significantly longer duration of symptoms, suggesting more advanced disease at the time of surgery. This difference could have influenced clinical outcomes and may, in part, account for the comparable results observed between the OT and VATS groups. Therefore, residual confounding due to differences in baseline disease severity cannot be completely excluded, even though this variable was adjusted for in the multivariable analysis.”

3. There are still typos throughout (e.g., “blood lose” should be “blood loss”).

Response: Thank you for highlighting the remaining typographical errors in our manuscript. We apologize for these oversights. We have carefully proofread the entire document and corrected all identified typos, including changing “blood lose” to “blood loss.” We have also performed a thorough language check to improve clarity and accuracy throughout the manuscript.

---

## [Editor Report · Decision Letter 2]

2 Sep 2025

Comparative Analysis of Surgical Outcomes: Video-Assisted Thoracoscopic Surgery versus Open Thoracotomy in Organizing Thoracic Empyema Management

PONE-D-25-37489R2

Dear Dr. Tantraworasin,

We’re pleased to inform you that your manuscript has been judged scientifically suitable for publication and will be formally accepted for publication once it meets all outstanding technical requirements.

Kind regards,

Luca Bertolaccini, M.D., Ph.D.

Academic Editor

PLOS ONE
---

## [Editor Report · Acceptance letter]

PONE-D-25-37489R2

PLOS ONE

Dear Dr. Tantraworasin,

I'm pleased to inform you that your manuscript has been deemed suitable for publication in PLOS ONE. Congratulations! Your manuscript is now being handed over to our production team.

Kind regards,

on behalf of

Dr. Luca Bertolaccini

Academic Editor

PLOS ONE